# Fabrication of Hydroxy-Terminated Polybutadiene with Piezoelectric Property by Functionalized Branch Chain Modification

**DOI:** 10.3390/molecules28041810

**Published:** 2023-02-14

**Authors:** Bo Yuan, Guang Wang, Wenxue Tian, Li Zhou, Chunxiang Li

**Affiliations:** 1Department of 206, Xi’an High-technology Institute, Xi’an 710025, China; 2The Forth Academy of CASC, Xi’an 710025, China; 3School of Chemistry and Chemical Engineering, Harbin Institute of Technology, Harbin 150001, China

**Keywords:** HTPB, branch chain modification, piezoelectric properties

## Abstract

Hydroxyl-terminated polybutadiene (HTPB)-based piezoelectric polymer (m-HTPB) is prepared for the first time by functionalized branch chain modification strategy. In the presence of HTPB with >98.8% cis-1,4 content, the C=C bond partly breaks down, and functionalized acetylferrocene groups are introduced to the cis-1,4 polybutadiene branch chain, retaining the high cis-1,4 content of HTPB. The whole process is conducted under mild conditions, without complicated manipulations. The microstructure and molecular weight of m-HTPB are characterized by Fourier-transform infrared (FTIR) spectra, ^1^H or ^13^C nuclear magnetic resonance spectrum (NMR), and gel permeation chromatography (GPC). The thermal properties of HTPB and m-HTPB are determined by differential scanning calorimetry (DSC). Electrochemical investigations reveal that m-HTPB exhibits higher conductance compared with HTPB. The m-HTPB flexible piezoelectric polymer is further used for in situ and real-time pressure monitoring. This simple and effective strategy provides a promising polymeric material for flexible piezoelectric sensors.

## 1. Introduction

With the continuous and rapid development in science and technology, the demand for flexible and wearable tactile sensors has increased due to their essential applications in wearable and implantable electronics and the human–machine interface in intelligent robots [1,2]. The tactile sensors are based on different mechanisms, such as piezoresistive [3], capacitive [4], and piezoelectric [5] sensing principles, and are used in different applications. Among these functional materials, piezoelectric materials generate electrical energy and signals when an external mechanical force is applied [6,7]; for example, MAPbI3 perovskite, which is a major concern for stretchable/flexible device applications, revealed a high piezoelectric coefficient of ~25 pm/V under illumination, and these kinds of organic–inorganic metal halide perovskites materials exhibit a structural phase transition similar to that of other inorganic perovskite materials, leading to ferroelectric polarization and piezoelectric properties [8]. The piezoelectric effect is a reversible process: piezoelectric materials can exhibit a reversible piezoelectric impact in such a way that when these materials are subjected to an electric field, they are capable of generating or producing mechanical strain. Thus, piezoelectric materials are frequently used as energy harvesters and self-powered sensors, owing to relatively easy manufacturing [9]. Moreover, the piezoelectric sensing mode provides a fast response to external loading, so as to promise great potential for developing completely autonomous wearable devices and systems.

Generally, piezoelectric materials are produced from crystals, ceramics, and polymers [10], such as lead zirconate titanate (PZT), Pb-free (Na_0.5_K_0.5_)NbO_3_, and poly (vinylidene fluoride) [11,12,13]. Although these lead-based and non-lead-based ceramic piezoelectric materials have good piezoelectric properties, the flexibility of such materials is generally poor, which limits their application in flexible pressure sensors. Many polymer-based piezoelectric materials, such as PVDF, possess good flexibility, piezoelectric properties, a dielectric constant, pyroelectric effect, ferroelectric effect, bio-compatibility, and chemical stability.

Polymers have many advantages over crystals or inorganic ceramics, including a light weight, higher flexibility, a relatively low piezoelectric coefficient, and dielectric permittivity, limiting the polarization mechanism of molecular dipoles [14]. Therefore, different strategies have been proposed for improving the polymers′ conductivity to facilitate signal transmissions, such as polymer modification and fillers by carbon-based materials, metal oxides, and metal particles [15]. For example, Gao et al. fabricated graphite-nanoplatelet-decorated polymer nanofiber composites with hierarchical structures by a combination of electrospinning and ultrasonication. The formation of a conductive percolation network in the nanofiber composite significantly improved the electrical conductivity of graphite-nanoplatelet-decorated polymer nanofiber composites [16]. They also prepared a flexible conductive polymer nanofiber composite by carbon nanotube decoration of the polyurethane/polyethersulfone nanofibers under the assistance of ultrasonication. The obtained composite mat possessed an ultralow percolation threshold of 0.056 vol%, and the electrical conductivity was as high as 2.8 S/m at a relatively low carbon nanotube concentration of 0.85 vol% [17]. Raza et al. dispersed the PbO nanoparticles in deionized water and mixed them with graphite to obtain graphite PbO (G-PbO) filler. Compared with pure polyvinyl chloride film, the conductivity of polyvinyl chloride (PVC)/G-PbO film was 42.65 times greater [18].

Hydroxy-terminated polybutadiene (HTPB) is a translucent liquid rubber that emerged in the 1960s, with good transparency and low glass transition temperature, high mechanical and thermal stability, and anti-aging properties [19,20]. In addition, it can be cured and cross-linked at room temperature or high temperature to form an elastomer with a three-dimensional network structure, which has excellent mechanical properties and good corrosion resistance, making it suitable as a solid propellant liner in the aerospace industry [21,22]. Thus, HTPB is widely employed as polyol to fabricate thermosetting polyurethanes, solid propellants, explosives, encapsulates, and gas separation membranes [23,24,25]. Since the properties of conjugated diene polymers mainly depend on their chain microstructure [26], the conventional polymerization process for HTPB can not well control their microstructure [27,28]; for instance, HTPB is usually prepared by free radical polymerization or anionic polymerization of 1,3-butadiene using a bifunctional initiator, in which the properties of products are influenced by reaction conditions, such as type of initiator, polarity of solvent, and polymerization temperature. With the development of organometallic catalysts, catalyzed chain growth, ring-opening metathesis polymerization, and acyclic diene metathesis polymerization in the presence of an appropriate chain transfer agent could be other choices to prepare telechelic polymers with terminal functional groups; however, the unrecoverable, expensive catalysts and harsh conditions have limited their applications. Moreover, the polyurethanes′ service life is reduced due to the formation of inevitable microcracks under external environmental stresses. The HTPB-based devices aggravate the crises of long service life demand and aging law, since HTPB as a raw material originates from petrochemical. Therefore, strategies to modify the molecular structure of HTPB are significant for sustainable industrial applications.

A conductive polymer has a unique conjugated structure (arranged by single and double bonds). Common conductive polymer materials are divided into intrinsic and conductive filler-filled polymers [29]. A filled conductive polymer is a multiphase composite exhibiting a certain conductivity that is formed by combining a nonconductive polymer with conductive particles via physical or chemical methods. Metal oxide nanoparticles are widely used as conductive fillers. Hamid et al. prepared PbO nanoparticles as conductive fillers to prepare nanocomposite polymer film (nCPM). With an increase in the PbO filler content from 5% to 25%, the conductivity of PVC-PbO-nCPM increased, reaching up to 0.002297 S/cm [30]. However, the conductivity decreased when the filler content was greater than 25%. In addition, polymers are also widely employed as conductive fillers. Hamid prepared three composite polymer membranes (CPMs) based on PVC and different fillers (sodium polyacrylate and sodium polyacrylate graphite) by the solvent casting method [31]. The conductivity of the CPMs was significantly higher than that of the other two, with the highest value of 0.00346 S/cm. However, the material’s conductivity decreased significantly when the filler exceeded 15%. A higher filler concentration caused surface aggregation, narrowing the ion channel and blocking the ions, and thus reducing the conductivity. The polymer′s conductivity when containing conductive filler is limited by the conductive filler, which is insufficient in economy and preparation technology. Unlike filled conductive polymers, intrinsic conductive polymers are formed by π conjugate structure design, creating a certain amount of conductive carriers for polymer molecules [32]. This strategy can convert the original insulating materials into conductive materials.

Herein, inspired by the designing strategy of intrinsic conductive polymer, the novel functionalized hydroxy-terminated polybutadiene (m-HTPB) is successfully fabricated with piezoelectric performance. To induce conductive properties in the produced m-HTPB, the unsaturated double bond was partly broken down by the addition of hydrogen chloride and functionalized acetylferrocene groups that were introduced by substitution reaction, contributing to the conjugated structure for cis-1,4 polybutadiene branch chain and retaining the high cis-1,4 content. This is the first time a conjugated structure has been incorporated into hydroxy-terminated polybutadiene. The resulting m-HTPB exhibits outstanding conductivity and piezoelectricity due to the high cis-1,4 structure in the polybutadiene main chain and the conjugated structure in the polybutadiene branched chain.

## 2. Results

### 2.1. Characterization of m-HTPB

Hydroxyl-terminated polybutadiene with a piezoelectric property (m-HTPB) was prepared via a two-step method by taking commercial butadiene rubber (BR900) as a starting material (Figure 1).

The molecular structures of BR-900 (purchased commercial-grade hydroxy-terminated polybutadiene), HTPB (prepared high-cis hydroxy-terminated polybutadiene), and m-HTPB (functionalized hydroxy-terminated polybutadiene) polymer were characterized by FT-IR (Figure 1). All the spectra show common peaks at ~729 cm^−1^ due to the cis-1,4 isomer, confirming the high cis-1,4 content in BR-900, HTPB, and m-HTPB [33]. Peaks at 3300~3500 cm^−1^ are attributed to the −OH symmetric stretching vibration in BR-900, HTPB, and m-HTPB [34]. The attachment of functionalized acetylferrocene with the polybutadiene is first confirmed by IR. The characteristic absorption bands at 556 and 1510 cm^−1^ are attributed to the elastic vibration absorption of the benzene ring skeleton and C-H vibration outside the plane, respectively. Moreover, the characteristic peaks at ~830, 1389, and 1560 cm^−1^ are assigned to the p-phenylene groups, Cp, and N-H stretching vibrations of functionalized acetylferrocene, respectively. The peaks at 1638 and 1406 cm^−1^ are attributed to the C=C stretching vibration of quinone and benzene rings, respectively, suggesting the existence of a conjugate structure in m-HTPB. In HTPB, the peak at ~729 cm^−1^ is due to cis-1,4 isomer, that at ~910 cm^−1^ is due to 1,2-vinyl structure, while that at ~965 cm^−1^ is due to trans-1,4 isomer. Figure 1 suggests that both HTPB and m-HTPB have cis-1,4 isomers.

The molecular structures of BR900, HTPB, and m-HTPB were further determined by NMR (Figure 2). The characteristic peak at 2.02 ppm is assigned to methylene (-CH_2_-) in trans -1,4 isomers, the peak at 4.77 ppm is due to the 1,2-vinyl structure, while that at 1.94 ppm for HTPB and m-HTPB is due to cis -1,4 isomers (Figure 2b) [27,28]. The peak area ratios for cis, trans -1,4 isomers and 1,2-vinyl structure are 98.6:0.3:1.1 for HTPB and 98.1:0.4:1.5 for m-HTPB, suggesting the cis content of 98.6% for HTPB and 98.1% for m-HTPB. The triplet peaks at 3.56~3.59 ppm are attributed to the methylene proton peak adjacent to the hydroxyl (-CH_2_OH). A clear peak due to the -CH_2_ (3.72 ppm) group indicates only one type of hydroxyl in HTPB and m-HTPB. In addition, the peak at 4.9 ppm corresponds to =CH_2_, while the =CH proton in cis-1,4 isomers appears at 5.38 ppm. The peaks around 5.38 ppm are dominant, suggesting that a high cis-1,4 content was retained in both HTPB and m-HTPB. The HTPB functionalization with functionalized acetylferrocene generated peaks at 3.3, 4.51, and 4.77 ppm, related to the benzene rings, acetyl meta hydrogen on the Cp ring, and acetyl ortho hydrogen on the Cp ring of functionalized acetylferrocene, respectively, indicating the complete reaction of functionalized acetylferrocene groups with HTPB. Some other peaks (5.31 ppm for proton in conjugated structure and 7.46 ppm for Ph-H) are also observed, confirming the reaction between HTPB and functionalized acetylferrocene groups. The peak at 2.41 ppm related to CH_2_ next to the double bond is observed for HTPB and m-HTPB, attributed to the polybutadiene structure, suggesting that the polybutadiene structure was retained during the two-step reaction.

In the ^13^C NMR spectra, the single peaks at 129.4 and 27.2 ppm correspond to =CH- and -CH_2_- carbon atoms in cis -1,4 isomers, while the peak at 32.5 ppm is assigned to the carbon atom for -CH_2_- in trans -1,4 isomers (Figure 3). Similarly, the -CH_2_OH characteristic peak in HTPB and m-HTPB appears at 62.6 ppm. However, the signals of the 1,2-vinyl structure (-CH=CH_2_, 142.3 ppm) are very weak for HTPB and m-HTPB, revealing that HTPB and m-HTPB with high cis-1,4 content were successfully fabricated. Comparing BR900 and HTPB, the new single peaks at 20.00 and 69.67 ppm in m-HTPB are assigned to carbon atoms of -CH_3_ and Cp [34], and the peaks at 139.7 and 201.7 correspond to carbon atoms in p-phenylenediamine and -CO- in functionalized acetylferrocene groups. The -CH_3_ group determines the substitution position in functionalized acetylferrocene and the molecular structure of m-HTPB.

^1^H NMR can measure the OH values; thus, 20 μL of styrene (internal standard) was added to the prepared rubber (12.3 mg) for ^1^H NMR spectra. To compare the integral area of the proton peak for =CH_2_ in styrene and −CH_2_OH in HTPB or m-HTPB (Figure 4), the OH values were calculated according to the following equation [27,28]:$$\frac{n_{St}}{n_{OH}}=\frac{n_{St}}{n_{{CH}_{2}}}=\frac{2\times [I\left(5.64-5.69\right)}{\left[I\left(3.56-3.59\right)\right]}$$

The OH values are 0.54 and 0.69 mmol/g for HTPB and m-HTPB, respectively.

Since the molecular weights of polymers obtained from GPC were larger than those calculated by ^1^H NMR due to polymer aggregation in THF [27,28], HTPB and m-HTPB were completely dissolved in this solvent. Thus, the molecular weight was further determined by GPC (Figure 5). The sharp GPC peaks for HTPB and m-HTPB indicate a narrow molecular weight distribution. From the retention time and standard curve equation (lg*Mn* = −0.919*x* + 12.007), the molecular weight of HTPB was 2596 g/mol, while that of m-HTPB was 3075 g/mol. By introducing functionalized acetylferrocene groups in the cis-1,4 polybutadiene branch chain, the molecular weight of the resulting m-HTPB increased gradually from 2596 to 3075 g/mol (Figure 5). The GPC traces show that the preparation of m-HTPB is feasible and highly efficient.

### 2.2. Thermal and Electrochemical Properties of m-HTPB

The glass transition temperatures (Tg) of BR900, HTPB, and m-HTPB were determined by DSC within the temperature range of −150 °C to 80 °C at a heating rate of 10 °C/min under a nitrogen atmosphere (Figure 6). The Tgs for BR900, HTPB, and m-HTPB were −106.2, −100.4, and −85.5 °C, respectively. Since the Tg of the polymer is greatly influenced by its chain microstructure, the Tg of −106.2 °C for BR900 is attributed to its flexible cis-1,4- polybutadiene chain. Compared with BR900, the Tg for HTPB was slightly higher due to the breakdown of the C=C bond and hydroxyl groups in the cis-1,4 polybutadiene chain ends. In addition, the melting peak for HTPB was lowered to ca. 57 °C, while BR900 melted at ca. −15 °C, due to the short polymer chain length and the functional groups, as evidenced by the OH value changes. In addition, the structure of m-HTPB was regular due to functionalized, acetylferrocene, branch-modified polymer chains, further increasing the Tg of m-HTPB. The disappearance of the melting peak indicates that the crystallization domain′s formation in m-HTPB with high cis- 1,4 content was weakened owing to their irregular microstructure.

Electrochemical impedance spectroscopy (EIS) was performed to investigate the conductivity of the polymer. Nyquist plots for HTPB and m-HTPB films in the frequency range of 10^−2^ Hz to 10^5^ Hz are shown in Figure 7. The suitable equivalent circuit is also exhibited in the inset according to the electrode//electrolyte//active layer//electrode device configuration. *C* is a double-layer capacitance on the surface of the active layer, and *W* is assigned to the starting linear part, which reflects a diffusive impedance of the Li^+^ ion entering into the electrodes. *R_S_* is a combinational resistance involving ionic resistance of electrolyte and contacting resistance at the active materials/ITO glass interface, which corresponds to the intercept of the Z’ axis. Different from the HTPB, the EIS for m-HTPB is a semicircle (charge transfer impedance at the electrode–electrolyte interface) in the high-frequency region and a straight line (ion diffusion process within the electrode) in the low-frequency region. The intercept of the semicircle in HTPB decreases from 1.54 × 10^5^ to 1.55 × 10^4^ in m-HTPB, implying a decrease in charge transfer resistance (R_CT_). The lower R_CT_ for m-HTPB is attributed to the π-electron-conjugated structure in functionalized acetylferrocene that improves the electron transport capacity. The enhanced electron conduction also benefits from the directly connected covalent bond between Cp and aniline. However, the influence of ion diffusion on the charge transfer is greater than that of electron conduction; thus, a high *R_CT_* was obtained due to the non-delocalized electron in HTPB.

### 2.3. Piezoelectric Performance of m-HTPB

When the sandwich structure was assembled with polydimethylsiloxane (PDMS) and m-HTPB as the outer and inner fluid, respectively, a vertical periodic compressive stress was applied to the as-prepared device, and the output current was measured by an electrochemical workstation for piezoelectric sensor testing. The compressive area was 0.5 cm^2^. The current output signals from m-HTPB were recorded, performed under a force at 1 Hz (Figure 8). Experimentally, the as-prepared device exhibited an output current peak of ca. 2.04 nA at 1 Hz (Figure 8a), indicating that the m-HTPB based device could convert the received pressure signal to an electrical signal and act as an energy-harvesting and self-powered sensing system. This m-HTPB-based sensor exhibited a fast response of 71 ms. The main challenge in piezoelectric composites is poor stress conduction since only a low-stress level is transferred to the piezoelectric functional layer. This stress is wasted when the composite material is subjected to external forces, causing poor piezoelectric conversion and output performance. Thus, it is important to investigate piezoelectric electronic performance under a small force. The output current peak of all the sensors increased step by step when the applied force was increased from 1 to 3 N, attributed to a higher piezoelectric response to a larger deformation of the piezoelectric m-HTPB polymer (Figure 8b). The current output of the sensor showed an approximative linear relationship with the applied force in the range of 1~3 N, demonstrating a sensitivity of 0.36 nA N*^−^*^1^. The short-circuit current of the device was measured under a periodic force at a magnitude of 1 N for the classic “switching polarity” tests. Figure 8b shows that the output current can be flipped with the same approximate values when the piezoelectric device is connected to the measurement system through the forward and reverse modes, indicating the piezoelectric response of m-HTPB.

## 3. Materials and Methods

### 3.1. Materials

Polybutadiene rubber (BR-900) was purchased from Shandong Jiyuan reagent Co. Ltd. BR-900 (cis-1,4 = 34.6%) was redissolved in cyclohexane (50−60 g/L), precipitated from ethanol to remove impurity, and then dried in vacuum at 50 °C overnight before being used. The solvents (tetrahydrofuran (THF), toluene, and ether) were purchased from XiLong Scientific Co., Ltd. All other chemicals were obtained from Aladdin and used as received, including 3-chloroperbenzoic acid (75%, Aladdin Chemistry Co., Ltd., Shanghai), periodic acid (H_5_IO_6_; 99%, Aladdin Chemistry Co., Ltd., Shanghai), Celite (CP, Sinopharm Chemical Reagent Co., Ltd., Shanghai), and Na_2_CO_3_ (CP, Sinopharm Chemical Reagent Co., Ltd., Shanghai).

### 3.2. Preparation of m-HTPB

An amount of 5.0 g BR-900 rubber was dissolved in 100 mL THF, and 3-chloroperoxybenzoic tetrahydrofuran solution was added dropwise to carry out epoxidation for 1.5 h at 30 °C. Then, periodic acid–tetrahydrofuran was added for an oxidative cracking reaction for 2 h. Then, certain amounts of sodium bicarbonate and 2,6-ditertbutyl-4-methylphenol antioxidant were added, and the mixture was kept overnight. Then, sodium borohydride in 50 mL THF was added and reacted at 30 °C for 2 h. The excess sodium borohydride was quenched by adding deionized water at room temperature. Then, 10 g Na_2_CO_3_ was added under stirring for 1 h. The mixture solution was spin evaporated to obtain HTPB with a yield of 88.5%. ^1^H-NMR (400 MHz, CDCl_3_): δ (ppm): 1.94 (s, 2 H), 2.02 (t, 2 H), 3.72 (s, 2 H), 5.38 (t, 2 H); ^13^C-NMR (125 MHz, CDCl_3_): δ (ppm): 27.2 (CH_2_), 32.5 (CH_2_), 62.6 (CH_2_OH), 129.4 (CH), 142.3 (CHCH_2_).

Before adding Na_2_CO_3_, 8 mL of concentrated hydrochloric acid was added dropwise to the mixture solution and reacted for 30 min. Then, 10 g Na_2_CO_3_ and a certain amount of functionalized acetylferrocene, which was prepared by refluxing acetylferrocene and p-phenylenediamine in methanol solution, were added and stirred for 1 h. The mixture solution was evaporated by spinning to obtain m-HTPB with a yield of 82.4%. ^1^H-NMR (400 MHz, CDCl_3_): δ (ppm): 1.43 (s, 3 H), 1.94 (s, 6 H), 2.02 (m, 2H), 3.72 (t, 2 H), 5.38 (t, 2 H); ^13^C-NMR (125 MHz, CDCl_3_): δ (ppm): 20.00 (CH_3_), 27.2 (CH_2_), 32.5 (CH_2_), 62.6 (CH_2_OH), 69.67 (Cp), 129.4 (CH), 139.7 (C_6_H_4_N_2_H_4_), 142.3 (CHCH_2_), 201.7 (CO).

### 3.3. Preparation of m-HTPB-Based Piezoelectric Sensor

An amount of 0.5 g of prepared m-HTPB was dissolved in 10 mL tetrahydrofuran, and the well-dispersed solution was poured into a mold. After evaporating the solvent at room temperature, the m-HTPB polymer film was dried in a vacuum oven at 70 °C overnight to remove solvent residue. Then, the m-HTPB polymer films were cut into pieces 1 cm × 0.5 cm.

The m-HTPB-based piezoelectric sensor was composed of polydimethylsiloxane (PDMS) decorated with copper foil conductive tape as top and bottom flexible packaging layers and m-HTPB polymer film as a piezoelectric layer. The top and bottom flexible packaging layers sandwiched an m-HTPB film to form a piezoelectric, self-powered tactile force sensor.

### 3.4. Characterizations

Fourier-transform infrared (IR) spectroscopy analysis of the samples was carried out on an FT-IR TENSOR27 Fourier-transform infrared spectrophotometer, and the diffuse reflectance spectra were scanned over the range of 600~4000 cm^−1^ (resolution = 2 cm^−1^, 100 scans per measurement). The samples were mixed with potassium bromide powder (≈4 mg sample/500 mg KBr), and background served as KBr powder. The molecular weight distributions of HTPB and m-HTPB were measured at 25 °C using a gel permeation chromatography system (1515–2414 series system, column: 300 × 7.5 mm) equipped with a Polymer Laboratories pump, a data stream refractive index detector, and an autosampler. The calibration was carried out using 12 polystyrene standards with *M_n_* values ranging from 162 to 371,100 (Varian Inc., UK, London). THF was used as an eluent, and the flow rate was 1.0 mL/min. The data were processed using Cirrus GPC offline GPC/SEC software (version 2.0). The microstructures of BR-900, HTPB, and m-HTPB were determined by ^1^H and ^13^C NMR spectra using a Bruker AVANCEIII-400 instrument (operated at 400.13 MHz (^1^H) and 125.77 MHz (^13^C)) in CDCl_3_ with styrene as the internal standard at room temperature. Differential scanning calorimetry (DSC) analysis was performed on a TA Q200 thermal analyzer under a nitrogen atmosphere. The samples were first cooled to −150 °C at 100 °C/min and then heated to 100 °C at 10 °C/min to remove the thermal history. After isothermalization at 100 °C for 3 min, the samples were subsequently cooled to −150 °C at 20 °C/min and reheated to 100 °C at 10 °C/min.

All electrochemical impedance spectroscopy tests were performed on the electrochemical analyzer (Dong Hua, DH-7000) with the three-electrode testing system. A platinum electrode (5 mm × 5 mm) was used as the counter electrode, and indium tin oxide (ITO) conductive glasses covered with HTPB or m-HTPB films were used as the working electrodes. An Ag/AgCl electrode was used as the reference electrode. The electrochemical impedance analysis of HTPB or m-HTPB film was performed in the frequency range from 10^−2^ to 10^5^ Hz with a signal amplitude of 5 mV in a 1 M H_2_SO_4_ solution. The piezoelectric performance tests were also completed by an electrochemical workstation. Experiments for mechanical test properties were performed using an Instron 5942. Stress experiments were performed at room temperature (25 °C). Each mechanical test was repeated with nine samples of each weight fraction by preparing three different batches of three samples.

## 4. Conclusions

Functionalized acetylferrocene branch chain modified hydroxy-terminated polybutadiene (m-HTPB) with a piezoelectric property was successfully prepared. In the addition/substitution reaction, the C=C bond partly broke, and functionalized acetylferrocene groups were introduced into the polybutadiene branch chain under mild conditions. The calculated cis-1,4 content for m-HTPB was higher than 98%. The functionalized acetylferrocene groups increased the glass transition temperatures from −100.4 °C to −85.5 °C due to their irregular microstructure. However, the conjugated structure of functionalized acetylferrocene groups improved the electron transport capacity and deeply reduced the charge transfer resistance for m-HTPB, which offers the piezoelectric property for m-HTPB. The as-prepared piezoelectric device with PDMS as the outer layer exhibited an approximative linear relationship to the applied vertical compressive stress. These results provide a promising HTPB modification strategy toward developing conductivity polymer and flexible piezoelectric sensors and may give some new implications for HTPB.

## Data Availability

Not applicable.

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
