# Peer review of "Fabrication of Hydroxy-Terminated Polybutadiene with Piezoelectric Property by Functionalized Branch Chain Modification"

_molecules, 2023, doi:10.3390/molecules28041810_

Round 1

Reviewer 1 Report

In this manuscript, Yuan et al. described “Functionalized Branch Chain Modification of Hydroxy-Terminated Polybutadiene with Piezoelectric Performance” in detail. Samples are properly characterized, and the activities are excellent. However, at this stage there are still many problems and I therefore suggest a major review for this manuscript keeping in mind the following questions.

1) The title of the manuscript has a problem; the respected authors are requested to write a precise title.

2) The English language is very poor. A definite relationship must exist between the different portions of the manuscript, for example abstract and conclusion are written in past tense, experimental part is written in past, result and discussion are written in present tense, while in introduction, shortcomings are written in present tense, review literature is written in past tense. No correlation is seen in the manuscript.

3) The introduction of the manuscript is not proper; the respected authors are requested to write a precise introduction.

4) The spectra of Nyquist plots are strange, please provide a fresh plat.

5) Any abbreviation must be defined completely before its first appearance but in abstract abbreviations such as “DSC” and “GPC”.

6) Some very important citations are missing.

i) A. Hamid, M. Khan, A. Hayat, J. Raza, A. Zada, A. Ullah, F. Raziq, T. Li, F. Hussain, Probing the physio-chemical appraisal of green synthesized PbO nanoparticles in PbO-PVC nanocomposite polymer membranes,    Spectrochim. Acta Mol. Biomol. Spectrosc. 235 (2020) 118303.

ii) J. Raza, A. Hamid, M. Khan, F. Hussain, L. Tiehu, P. Fazil, A. Zada, Z. Wahab, A. Ali, Spectroscopic characterization of biosynthesized lead oxide (PbO) nanoparticles and their applications in PVC/graphite-PbO nanocomposites, Z. Phys. Chem. 236 (2022) 619-636.

iii) A. Hamid, M. Khan, F. Hussain, A. Zada, T. Li, D. Alei, A. Ali, Synthesis and physiochemical performances of PVC-sodium polyacrylate and PVC-sodium polyacrylate-graphite composite polymer membrane, Z. Phys. Chem. 235:12 (2021) 1791-1810.

7) The conclusion requires careful attention in terms of findings and conclusions.

8) Many sentences are too long and meaningless, for example “Hydroxy-terminated polybutadiene (HTPB) is a liquid rubber, and due to its excellent physical properties [18-19], such as low viscosity, low glass transition temperature [20-21], high mechanical and thermal stability, better thermodynamic and anti-aging properties, it is widely employed as polyol to fabricate thermosetting polyurethanes (PUs).” Please write short and meaningful sentences.

Author Response

Dear Editor:

Thank you very much for your kind letter, along with the constructive comments of the reviewer concerning our manuscript (molecules-2189260). We have thoroughly considered all the comments of the reviewers and substantially revised our manuscript, and the major revised portions are marked in yellow in our revised manuscript (please see the attachment). We also respond point by point to the reviewer’s comments as listed below, along with a clear indication of the location of the revision. We look forward to hearing from you.
Thanks again
Sincerely yours

Chunxiang Li

Response to Reviews

For Reviewer #1:

Question 1:

The title of the manuscript has a problem; the respected authors are requested to write a precise title.

Answer:

We are grateful to the reviewer 1 for his suggestion. As you suggested, we rephrased the title of the manuscript.

Question 2:

The English language is very poor. A definite relationship must exist between the different portions of the manuscript, for example abstract and conclusion are written in past tense, experimental part is written in past, result and discussion are written in present tense, while in introduction, shortcomings are written in present tense, review literature is written in past tense. No correlation is seen in the manuscript.

Answer:

Thanks for the suggestion from reviewer 1, we carefully checked the English in the text and the mistakes including some grammar and spelling errors all over the manuscript.

Question 3:

The introduction of the manuscript is not proper; the respected authors are requested to write a precise introduction.

Answer:

As you suggested, we rephrased the introduction part in the revised manuscript  (page 1: 41-44 line; page 2: 55-71 line).

Question 4:

The spectra of Nyquist plots are strange, please provide a fresh plat.

Answer:

As you suggested, we redraw the spectra of Nyquist in a fresh plat in the revised manuscript.

Question 5:

Any abbreviation must be defined completely before its first appearance but in abstract abbreviations such as “DSC” and “GPC”.

Answer:

Thanks for the suggestion from reviewer 1. We added the complete definition before abbreviation for “DSC” and “GPC” in the revised manuscript (page 1: 15-18 line).

Question 6:

Some very important citations are missing.

  1. i) A. Hamid, M. Khan, A. Hayat, J. Raza, A. Zada, A. Ullah, F. Raziq, T. Li, F. Hussain, Probing the physio-chemical appraisal of green synthesized PbO nanoparticles in PbO-PVC nanocomposite polymer membranes,    Spectrochim. Acta Mol. Biomol. Spectrosc. 235 (2020) 118303.
  2. ii) J. Raza, A. Hamid, M. Khan, F. Hussain, L. Tiehu, P. Fazil, A. Zada, Z. Wahab, A. Ali, Spectroscopic characterization of biosynthesized lead oxide (PbO) nanoparticles and their applications in PVC/graphite-PbO nanocomposites, Z. Phys. Chem. 236 (2022) 619-636.

iii) A. Hamid, M. Khan, F. Hussain, A. Zada, T. Li, D. Alei, A. Ali, Synthesis and physiochemical performances of PVC-sodium polyacrylate and PVC-sodium polyacrylate-graphite composite polymer membrane, Z. Phys. Chem. 235:12 (2021) 1791-1810.

Answer:

We are grateful to the Reviewer 1 for his suggestion. We cited all the suggested related works to increase the research significance and depth of our work.

Question 7:

The conclusion requires careful attention in terms of findings and conclusions.

Answer:

As you suggested, we rephrased the conclusion part in the revised manuscript (page 10: 296-301 line).

Question 8:

Many sentences are too long and meaningless, for example “Hydroxy-terminated polybutadiene (HTPB) is a liquid rubber, and due to its excellent physical properties [18-19], such as low viscosity, low glass transition temperature [20-21], high mechanical and thermal stability, better thermodynamic and anti-aging properties, it is widely employed as polyol to fabricate thermosetting polyurethanes (PUs).” Please write short and meaningful sentences.

Answer:

As you suggested, we have shorted many too long and meaningless sentences in the revised manuscript.

Reviewer 2 Report

Article comments
This is an interesting study on the Functionalized Branch Chain Modification of Hydroxy-Terminated
Polybutadiene with Piezoelectric Performance. The research was conducted properly. The
manuscript may be considered for its publication after addressing following comments.
The author(s) are suggested to kindly revise the manuscript for English language usages. For
example, Page 2, line 47, change “this” by “these”.
Instead of writing “cis” authors should write “cis” in all manuscript.
Page 2, line 72, HTPB and m-HTPB refer to modified and unmodified PB, respectively, but what type
of material is BR-900? At the beginning of the results section, the authors should indicate the
meaning of HTPB, m-HTPB, and BR-900, as well as the structure of these polymers.
Page 2, line 73: “Both” refers to two spectra, but in Fig. 1 appear three of them.
Line 74: The cis-1,4 content of HTPB cannot be determined by FTIR. This is not a quantitative
technique. If the autor(s) made any consideration for this technique to become quantitative, this
should be clearly stated.
Page 3, lines 83-87: Determining the composition of a polymer by integrating the area of the FTIR
peaks is not correct. Moreover, in the spectrum of HTPB (Figure 1), the peaks used to determine the
composition are not clearly appreciated.
Page 3, line 95: the proton for the terminal methylene (=CH2) always appears below 5 ppm, at about
4.9 ppm.
Page 4, lines 117 and 119: the experiments was conducted in THF, THF-d or CDCl3?
The spectra in Figure 2 are not clear, the signals overlap. The authors should improve this Figure
considering the scale and the solvent signal (this is very strong). Also, expansions could be shown.
Why did not the authors use 1H NMR spectra to determine the cis-1,4, trans-1,4, and 1,2 content of
HTPB?
To enhance the information on telechelic oligomers I suggest reviewing and including the reference:
Polymer Degradation and Stability 166 (2019) 202-212.
https://doi.org/10.1016/j.polymdegradstab.2019.05.021

Author Response

Dear Editor:

Thank you very much for your kind letter, along with the constructive comments of the reviewer concerning our manuscript (molecules-2189260). We have thoroughly considered all the comments of the reviewers and substantially revised our manuscript, and the major revised portions are marked in yellow in our revised manuscript (please see the attachment). We also respond point by point to the reviewer’s comments as listed below, along with a clear indication of the location of the revision. We look forward to hearing from you.
Thanks again
Sincerely yours

Chunxiang Li

Response to Reviews

For Reviewer #2:

Question 1:

The author(s) are suggested to kindly revise the manuscript for English language usages. For example, Page 2, line 47, change “this” by “these”. Instead of writing “cis” authors should write “cis” in all manuscript.

Answer:

We are grateful to the Reviewer 2 for his suggestion. We have carefully checked the English in the text and the mistakes including some grammar and spelling errors also.

Question 2:

 Page 2, line 72, HTPB and m-HTPB refer to modified and unmodified PB, respectively, but what type of material is BR-900? At the beginning of the results section, the authors should indicate the meaning of HTPB, m-HTPB, and BR-900, as well as the structure of these polymers.

Answer:

This comment raised by reviewer 2 is of great significance. We supplemented the definition of BR-900, HTPB and m-HTPB at the beginning of the results section, as well as the structure of these polymers (page 2: 88-90 line; page 3: 93-96 line).

Question 3:

 Page 2, line 73: “Both” refers to two spectra, but in Fig. 1 appear three of them.

Answer:

Thanks for the suggestion from reviewer 2. We have changed “Both” into “All” in the revised manuscript.

Question 4:

 Line 74: The cis-1,4 content of HTPB cannot be determined by FTIR. This is not a quantitative technique. If the autor(s) made any consideration for this technique to become quantitative, this should be clearly stated.

Answer:

As you suggested, we deleted inaccurate analysis and description in the revised manuscript.

Question 5:

Page 3, lines 83-87: Determining the composition of a polymer by integrating the area of the FTIR peaks is not correct. Moreover, in the spectrum of HTPB (Figure 1), the peaks used to determine the composition are not clearly appreciated.

Answer:

This comment from reviewer 2 is very relevant. We deleted inaccurate analysis and description for the composition determining of polymer by integrating the area of the FTIR peaks in the revised manuscript. For Fig. 1, we added some detailed description into that part to clearly show the composition of polymer, so we rephrased the explanation to improve this part (page 3: 99-104 line). 

Question 6:

 Page 3, line 95: the proton for the terminal methylene (=CH2) always appears below 5 ppm, at about 4.9 ppm.

Answer:

Thanks for the suggestion from reviewer 2. We have changed 5 into 4.9 in the revised manuscript.

Question 8:

Page 4, lines 117 and 119: the experiments was conducted in THF, THF-d or CDCl3?

Answer:

For NMR measurments, all the experiments were conducted in CDCl3. Thus, we rephrased the explanation to improve this part (page 5: 157-159 line).

Question 9:

 The spectra in Figure 2 are not clear, the signals overlap. The authors should improve this Figure considering the scale and the solvent signal (this is very strong). Also, expansions could be shown.

Answer:

As you suggested, we re-did and re-draw the Fig. 2, we added some detailed description into that part to clearly show the structure of m-HTPB (page 4: 116-123 line; page 4: 126-133 line).

.

Question 10:

 The conclusion requires careful attention in terms of findings and conclusions.

Answer:

We are grateful for the suggestion from reviewer 2. As you suggested, we rephrased the conclusion part in the revised manuscript (page 10: 296-301 line).

.

Question 11:

 Why did not the authors use 1H NMR spectra to determine the cis-1,4, trans-1,4, and 1,2 content of HTPB?

Answer:

Inspired by the reviewers’ question, we added description for cis-1,4 content of HTPB based on 1H NMR results, and rephrased the explanation to improve this parts (page 4: 116-120 line).

Question 12:

To enhance the information on telechelic oligomers I suggest reviewing and including the reference: Polymer Degradation and Stability 166 (2019) 202-212. https://doi.org/10.1016/j.polymdegradstab.2019.05.021

Answer:

We are grateful to the Reviewer 2 for his suggestion. We cited all the suggested related works to increase the research significance and depth of our work.

Reviewer 3 Report

molecules-2189260 

The article: “Functionalized Branch Chain Modification of Hydroxy-Terminated Polybutadiene with Piezoelectric Performance” by B. Yuan et al. describes the chemical modifications of a commercially high 1,4-cis polybutadiene rubber (BR900) introducing acetylferrocene groups, increasing by this way its conductivity after electrochemical investigations. The authors also prepared a piezoelectric device with the modified PB polymer and PDMS as the outer layer. Although the topic and the results seem interesting and promising for future applications as piezoelectric sensors, the manuscript is not well-written and presented. After careful reading, several comments were raised, and I feel that important information is absent from the manuscript. I suggest that if the authors address the following comments, it might be considered for publishing in Molecules.   

·       I was surprised that there were not any reactions in the manuscript. The authors discuss a lot of chemical modifications in the manuscript, but not a single reaction is shown. I believe it is necessary to add a scheme with all the chemical transformations the author performed.

·       Figure 1: I am not sure if the peak that the authors claim that corresponds to the Fe-C in the blue spectrum is accurate or it can be attributed to noise. It is not clear.

·        Figure 2: What is the high peak attributed to next the –CH2OH peak (3.7-3.8 ppm)?   

In general, it will help the reader if the authors provide the structures of the modified polymers and to assign letters or numbers to the protons thet they show in the spectra.

·       Regarding the same figure (Fig. 2): The authors claim “The functionalization of HTPB with functionalized acetylferrocene resulted in the appearance of the peaks at 2.41, 98 3.3, 4.51, 4.77 and 6.55 ppm…”. Actually, some of these peaks were also appeared in the HTPB precursor, i.e. the 2.41 and 6.55 ppm, respectively. Could the authors comment on this? Additionally, the authors write: “Some other new peaks appeared, confirming the reaction between HTPB and functionalized acetylferrocene groups.” (Line 102-103). Which are these “other peaks”?

·       The authors should mention the yield of each functionalization step and prove it through NMR spectroscopy.

·       Line 127-128: Is there any reference to support this assumption? Is that the reason that GPC shows higher MW than the NMR? What about the hydrodynamic volume of PB? Furthermore, could the authors provide some details in the materials and methods section on how many columns their GPC is equipped? It seems strange that such a low MW polymer (they write 2596 g/mol for the HTPB) is eluted between 9-10 min in the GPC trace. Also no information concerning the dispersity of the functionalized polymers. What is the Ð of each precursor?

Minor comments:

·       I suggest the Materials and Methods section to be placed before the Results.

·       Line 217: What is HIT stands for?

·       English editing of the manuscript is required.

Author Response

Dear Editor:

Thank you very much for your kind letter, along with the constructive comments of the reviewer concerning our manuscript (molecules-2189260). We have thoroughly considered all the comments of the reviewers and substantially revised our manuscript, and the major revised portions are marked in yellow in our revised manuscript (please see the attachment). We also respond point by point to the reviewer’s comments as listed below, along with a clear indication of the location of the revision. We look forward to hearing from you.
Thanks again
Sincerely yours

Chunxiang Li

Response to Reviews

For Reviewer #3:

Question 1:

 I was surprised that there were not any reactions in the manuscript. The authors discuss a lot of chemical modifications in the manuscript, but not a single reaction is shown. I believe it is necessary to add a scheme with all the chemical transformations the author performed.

Answer:

We are grateful for the suggestion from reviewer 3. As you suggested, we added a scheme for the chemical transformations (Scheme 1) to make the results more intuitively (page 3).

Question 2:

 Figure 1: I am not sure if the peak that the authors claim that corresponds to the Fe-C in the blue spectrum is accurate or it can be attributed to noise. It is not clear.

Answer:

I agree with this comment from reviewer 3, it is not fully suitable to label Fe-C there to explain our experiments. So we deleted unreasonable marks and descriptions in the revised manuscript (page 3: 99-104 line). 

Question 3:

  Figure 2: What is the high peak attributed to next the –CH2OH peak (3.7-3.8 ppm)?

Answer:

Inspired by the reviewers’ suggestion, we added some expatiation for the peak at 3.72 ppm. So we rephrased the explanation to improve this part. (page 4: 122-123 line)

Question 4:

 In general, it will help the reader if the authors provide the structures of the modified polymers and to assign letters or numbers to the protons thet they show in the spectra.

Answer:

We are grateful for the suggestion from reviewer 3. As you suggested, we added the structures of the modified polymers and to assign letters to the protons in Fig. 2.

Question 5:

 Regarding the same figure (Fig. 2): The authors claim “The functionalization of HTPB with functionalized acetylferrocene resulted in the appearance of the peaks at 2.41, 98 3.3, 4.51, 4.77 and 6.55 ppm…”. Actually, some of these peaks were also appeared in the HTPB precursor, i.e. the 2.41 and 6.55 ppm, respectively. Could the authors comment on this? Additionally, the authors write: “Some other new peaks appeared, confirming the reaction between HTPB and functionalized acetylferrocene groups.” (Line 102-103). Which are these “other peaks”?

Answer:

This comment raised by reviewer 3 is of great significance. we added some detailed description into that part to clearly show the results of 1H NMR, so we rephrased the explanation to improve these parts (page 4: 116-123 line; 126-135 line). 

Question 6:

 The authors should mention the yield of each functionalization step and prove it through NMR spectroscopy.

Answer:

As you suggested, we added the yield of each functionalization step and NMR spectroscopy results in the experimental section (page 8: 242-249 line; page 9: 250-254 line).

Question 7:

 Line 127-128: Is there any reference to support this assumption? Is that the reason that GPC shows higher MW than the NMR? What about the hydrodynamic volume of PB? Furthermore, could the authors provide some details in the materials and methods section on how many columns their GPC is equipped? It seems strange that such a low MW polymer (they write 2596 g/mol for the HTPB) is eluted between 9-10 min in the GPC trace. Also no information concerning the dispersity of the functionalized polymers. What is the Ð of each precursor?

Answer:

We are grateful to the Reviewer 3 for his suggestion. We added the related reference and rephrased a detailed description for Materials and Methods section (page 5: 149-150 line; page 9: 267-273 line). Besides, we added some description for the dispersity of the functionalized polymers to improve these parts (page 5: 157-159line)

Question 8:

  I suggest the Materials and Methods section to be placed before the Results.

Answer:

Based on the requirements of template, we are very sorry that we cannot change the position of the Materials and Methods section and the Results. For the convenience of reading, we added a brief description including structure and synthesis method for m-HTPB before the discussion of the results.

Question 9:

  Line 217: What is HIT stands for?

Answer:

As you suggested, we deleted this confusing description in the revised manuscript.

Round 2

Reviewer 1 Report

Since the authors made significant changes and improved the quality of the paper by responding well to all my questions and carrying out necessary changes, I therefore accept the publication of this paper in your reputed journal.

Reviewer 3 Report

The authors have answered in a satisfactory way to all of my queries.

I suggest that it can be published in its current form.